Review  

Subject Area:
cellular biology

Keywords:
breast implant-associated anaplastic large cell lymphoma, anaplastic large cell lymphoma, breast implants

Author for correspondence:
Lukas Kenner
e-mail: lukas.kenner@medunwien.ac.at

# Is breast implant-associated anaplastic large cell lymphoma a hazard of breast implant surgery?

Florian Fitzal[1], Suzanne D. Turner[2,6] and Lukas Kenner[3,4,5,6,7]

[1]Department of Surgery and Comprehensive Cancer Center, Medical University Vienna, Vienna, Austria
[2]Division of Cellular and Molecular Pathology, Department of Pathology, University of Cambridge, Cambridge CB20QQ, UK
[3]Ludwig Boltzmann Institute for Cancer Research, 1090 Vienna, Austria
[4]Unit of Laboratory Animal Pathology, University of Veterinary Medicine Vienna, 1210 Vienna, Austria
[5]Division of Experimental Pathology, and [6]Department for Experimental and Laboratory Animal Pathology, Clinical Institute of Pathology, Medical University of Vienna, 1090 Vienna, Austria
[7]The European Research Initiative for ALK-related Malignancies (ERIA), Cambridge, UK

SDT, 0000-0002-8439-4507; LK, 0000-0003-2184-1338

Breast implant-associated anaplastic large cell lymphoma (BIA-ALCL) may occur after reconstructive or aesthetic breast surgery. Worldwide, approximately 1.7 million breast implant surgeries are performed each year. To date, over 500 cases of BIA-ALCL have been reported around the world, with 16 women having died. This review highlights the most important facts surrounding BIA-ALCL. There is no consensus regarding the true incidence rate of BIA-ALCL as it varies between countries, is probably significantly under-reported and is difficult to estimate due to the true number of breast prostheses used largely being unknown. BIA-ALCL develops in the breast mostly as a seroma surrounding the implant, but contained within the fibrous capsule, or more rarely as a solid mass that can become invasive infiltrating the chest wall and muscle, in some instances spreading to adjacent lymph nodes, in these cases having a far worse prognosis. The causation of BIA-ALCL remains to be established, but it has been proposed that chronic infection and/or implant toxins may be involved. What is clear is that complete capsulectomy is required for treatment of BIA-ALCL, which for early-stage disease leads to cure, whereas chemotherapy is needed for advanced-stage disease, whereby improved results have been reported with the use of brentuximab. A worldwide database for BIA-ALCL and implants should be supported by local governments.

## 1. BIA-ALCL: an uncommon yet distinct disease entity

Breast implants have been the backbone of reconstructive and aesthetic breast surgery for many years, and have had a significant impact on patients' psychological well-being. Produced from apparently inert, non-toxic materials, breast implants have largely been considered safe and relatively risk-free, although concerns have been raised in the past [1–3]. However, a distinct group of lymphomas are increasingly being reported in association with breast implants. Breast implant-associated anaplastic large cell lymphoma (BIA-ALCL) is a form of non-Hodgkin T-cell lymphoma (NHL) and presents as an accumulation of T-lymphocytes, presenting as a late-onset seroma in one breast, with high CD30 expression and an absence of anaplastic lymphoma kinase (ALK) expression (figure 1). By contrast, non-implant-associated NHL found in the breast are mostly B-cell lymphomas [4].

The first case of BIA-ALCL was reported in 1997 in association with implants in a woman's breast [5]. Since 1997, many case reports have been published in the medical literature, although it was not until 2008 that the first series of cases

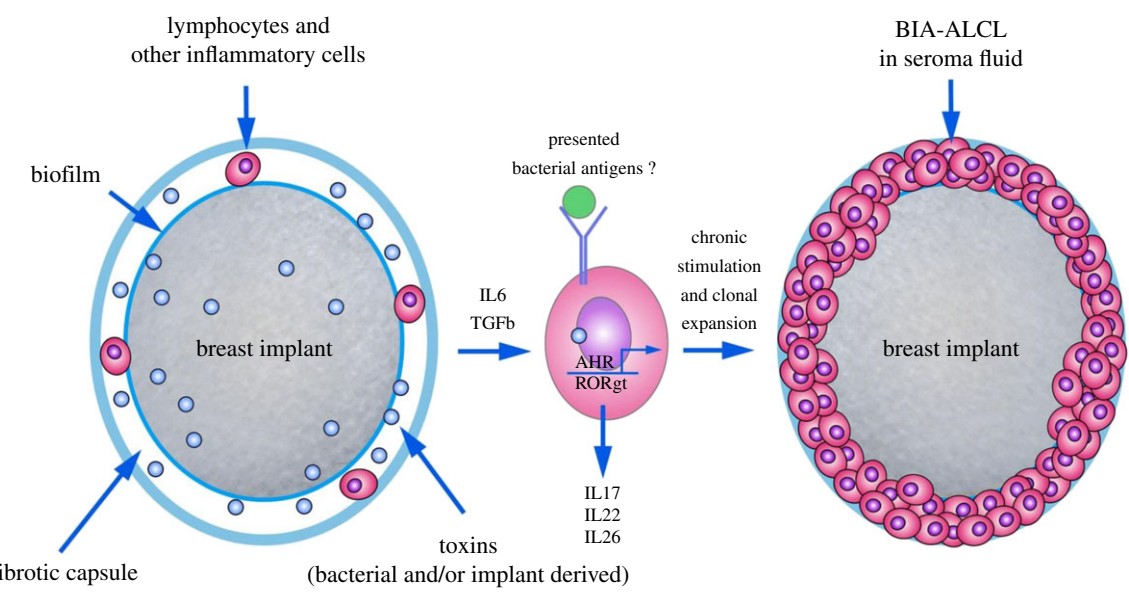

**Figure 1.** The pathogenesis of BIA-ALCL is probably multifactorial but probably involves some form of chronic stimulus provided in either an antigen-dependent (potentially via bacterial antigens) or antigen-independent manner (via cytokine stimulation and/or toxins activating the aryl hydrocarbon receptor) through which mutations are acquired including those involving the JAK/STAT pathway.

($n = 11$) was published by Daphne de Jong, a pathologist at the Netherlands Cancer Institute, Amsterdam [6]. In this series, cases had been diagnosed between 1994 and 2006 in women with a median age of 40 and all having surgery for cosmetic reasons. However, it was not until 2011 that an official statement from the US Food and Drug Administration (FDA) was published. They reported on BIA-ALCL in the USA [7]. At that time, the FDA had knowledge of 63 cases worldwide, but with a total number of breast implant procedures worldwide estimated at 5–10 million, the risk was deemed extremely rare [8]. As such, in 2011, the FDA stated that 'it was not possible to confirm with statistical certainty that breast implants cause ALCL'. Since 2011, awareness has improved, although slowly, with more case series of BIA-ALCL being reported, resulting in the provisional designation of BIA-ALCL as a distinct clinical entity in the 2016 World Health Organization (WHO) classification of lymphoid neoplasms, and the acceptance of a link between breast implants and ALCL by the FDA 1 year later, with 414 Medical Device Reports of BIA-ALCL and nine attributable deaths [9,10].

The FDA change in status on BIA-ALCL between 2011 and 2017 has been mediated by a number of BIA-ALCL series reports from across the world. The first larger population-based analysis from the Netherlands, published in 2008 described 389 women with breast lymphomas over a period of 17 years (1990–2006; population of 8 million). Within this series, 11 (3%) had histologically confirmed diagnoses of ALCL, of which five were associated with breast implants ($5/11 = 45\%$) [6]. The largest series published to date was of 173 histologically confirmed cases gleaned from the worldwide literature demonstrating that every woman had received a textured implant as some point in their clinical history [11]. A later review by Srinivasa *et al.* [11] investigated published incidences of BIA-ALCL and found 363 reported cases including 258 entries in the U.S. Manufacturer and User Facility Device Experience (MAUDE) database, although only 130 of these were histologically confirmed as BIA-ALCL. More recently, country-specific incidences have been published: 72 in Australia (of which three were lethal) [12], 41 cases in the UK [13,14],

22 cases in Italy [15], 43 in the Netherlands [16], 19 in France [4], seven cases in Germany [17] and 149 confirmed cases in the USA (414 reported overall including nine deaths) [10,18].

However, all of these studies probably represent an unclear estimation of actual incidence due to the number of unreported breast implant surgeries and the incomplete recording of the incidence of BIA-ALCL. Currently, the risk of a woman developing BIA-ALCL after undergoing implant-based surgery is largely unknown, with reported incidences varying from country to country. Various health experts such as the American Society of Plastic Surgeons, the MD Anderson Cancer Center or the Australian Government Department of Health state that based on currently available data, it is not possible to provide an accurate estimate of risk. Current expert opinion puts the risk of BIA-ALCL at 1 : 2832 for women with polyurethane implants in Australia and New Zealand, and 1 : 30 000 for women with any textured implant in the USA, with the current UK risk estimated to be 1 : 24 000, a figure based on all breast implants sold [13,18,19].

## 2. BIA-ALCL: diagnosis and therapy

ALCL can present as one of four types: systemic ALCL, which is either ALK-positive (largely paediatric) or ALK-negative, cutaneous ALCL, and the relatively recently described BIA-ALCL [9,20,21]. All share some key features such as positivity for CD30 expression but differ in prognosis and clinical course. While largely indolent, BIA-ALCL may rarely be aggressive and lead to death, as has been reported for 12 women so far [22,23]. In almost all reported cases, late-onset seromas surrounding the breast implants (median 8 [1–36] years of inserting the breast implants) are seen, with key symptoms including capsule thickening and capsular contracture characteristically manifesting as unilateral breast enlargement [22]. Late-onset seromas (greater than 1 year) should be investigated by ultrasound of the breasts, chest wall and regional lymph nodes. When ultrasound is inconclusive, magnetic resonance imaging or computed tomography scans may assist with the diagnosis of soft tissue

royalsocietypublishing.org/journal/rsob    Open Biol. **9**: 190006

masses [24]. Ultrasound-guided aspiration of seroma fluid with subsequent cytology, flow cytometry and analysis of the cells, primarily for CD30 expression and negativity for ALK helps to establish the diagnosis [25]. Patients may also present with clinical symptoms such as dermatitis of the breast and reactive enlarged axillary lymph nodes, in such cases, excision biopsies followed by histopathological analyses can aid in diagnosis [25].

Cases diagnosed as seroma are sometimes curable by surgery alone (stage 1A; complete removal of the implant and the surrounding fibrous capsule; a video of the removal of an implant with BIA-ALCL can be seen at http://links.lww.com/PRS/C360, although some also require adjuvant chemotherapy (stage IIA) [22,26]. However, the best oncological result can only be achieved by surgical treatment aiming for complete surgical excision [26]. Radiotherapy is indicated on occasions when there are positive margins on excision, unresectable or residual disease [25]. If tumours are non-resectable and have spread to the breast parenchyma, chest wall and/or lymph nodes, chemotherapy is indicated [25]. At present, chemotherapy takes the form of the standard T-cell lymphoma regimen, CHOP, with varying rates of success [15]. The moderate amount of anecdotal data produced so far suggests that event-free survival is far superior on treatment with the anti-CD30 drug conjugate brentuximab, which is unfortunately not available in all countries through National Healthcare [27,28].

## 3. Causes of BIA-ALCL

At present, the aetiology of BIA-ALCL is unclear, but the consistent association of breast implants with this distinct tumour entity lends some clues [20,29]. While considered relatively inert, the full extent of potential *in vivo* toxicity of implant materials, leachables and particulates is not completely understood. In addition, while breast surgery is considered a 'clean contaminated site', infection at the site of the implant may be a contributing factor; indeed, some other NHL are associated with chronic infection [30]. In addition, clusters of women presenting with BIA-ALCL have been reported consistent with an infectious aetiology [19] . Finally, due to the rarity of this disease and the lack of reported cases in some populations (such as Asian women), there is likely to be a genetic element to this disease [11]. Breast implants have been through many iterations over the years; at present, the fifth generation of implants are available, designed with the intention of reducing the rate of capsular contracture and preventing rotation, while providing an anatomically pleasing shape [31,32]. Such features have been provided by implants with textured surfaces, which are attributable to the large majority of BIA-ALCL cases [19]. When comparing the number of textured implants involved in BIA-ALCL diagnoses to the number of non-textured implants, the former clearly make up the majority (table 1). However, one must not forget that the type of implant was known in only half of the reported cases, thus rendering interpretation of these data difficult. An FDA report has identified 30 cases of BIA-ALCL associated with a smooth implant surface (https://www.fda.gov/Medical Devices/ProductsandMedicalProcedures/ImplantsandPros thetics/BreastImplants/ucm481899.htm), but the clinical histories for women are in some cases unknown with most

shown to have received a textured implant at some point in the past. Indeed, a recent report from Australia and New Zealand provided evidence that in 55 cases of BIA-ALCL, all women had at some point a textured prosthesis implanted [19]. There are differences in texture possibly leading to different BIA-ALCL risk [19,35].

As the majority of cases are diagnosed in the context of a textured implant, or history of a textured implant, this raises the question as to the reasons for this association. Increased T-cell infiltration in textured implants has been reported due to bacteria-colonized biofilms enveloping the prosthesis, and infections are associated with an elevated rate of seroma formation and capsular contraction [36]. Chronic infection is considered in some types of human lymphomas to be the causing mechanism, and in many others a contributing process [37]. It has, for instance, been known for a long time that *Helicobacter pylori* is a leading cause for inflammation-driven gastric cancer and mucosal-associated lymphoid tissue (MALT) lymphoma [30]; antibiotic therapy can indeed result in disease remission in some cases. Indeed, the role of the microbiome in a number of cancers including breast, oral, prostate and colorectal is becoming more apparent [38]. An infectious aetiology has not been reported for ALCL, although a rare number of systemic ALCL have been attributed to insect bites [20,39,40]. Whether or not chronic infectious agents/an inflammatory microenvironment drive the genetic events in ALCL is also not known, although *Ralstonia* sp. found in biofilms forming on the implant have been implicated in BIA-ALCL [41]. Moreover, the co-expression of CD30, a protein essential in the diagnosis of ALCL and normally found on activated T cells, further hints towards activated T cells being the origin of this disease. Chronic, T-lymphocyte-driven, but also fibroblast- and macrophage-associated inflammation induces the formation of a capsule that surrounds all implants. Whether the link between BIA-ALCL and breast implants is due to bacterial infection mediated by textured surfaces remains to be emphatically shown, but Hu *et al.* showed a difference in the bacterial composition of the biofilm between involved and uninvolved prostheses in women diagnosed with BIA-ALCL suggestive of an association, although this study only examined three patient's implants in this regard [41]. In further evidence, by adopting a peri-operative 14-point plan aiming to reduce biofilm formation, eight experts succeeded in furnishing evidence, through a retrospective analysis of their 21 000 patients with textured implants, that not a single case of BIA-ALCL occurred after a median observation period of 11.7 years, although the methodologies used in this study are debatable [19,42]. Hence, while there is no definitive proof of an association between BIA-ALCL and infection, the risk of infection should nevertheless be prevented as a matter of good surgical practice. General proposals for a reduction in the risk of infections caused by the implant, and the above-mentioned 14-point plan have been presented in this vein [36]. The latter consists essentially of peri-operative and localized antibiotic treatment, short-time exposure of the implant to air, the prevention of skin-implant contact and the omission of any drainage tubes.

A large number of chemically active agents from metals to vinyl derivatives and a number of aromatic hydrocarbons including benzol and xylol have consequently been detected in implant gels and exudates [43–45]. As it has been proven that the main components of the implants (e.g. silicones) may

royalsocietypublishing.org/journal/rsob    Open Biol. **9**: 190006

royalsocietypublishing.org/journal/rsob Open Biol. 9: 190006

**Table 1.** Published studies reporting cases of BIA-ALCL, their incidences, association with implant surface texture, length of clinical follow-up and clinical features. nr, not reported.

| region | number of cases | prosthesis surface | | | follow-up (months) | deaths | presented with | | ref. |
|---|---|---|---|---|---|---|---|---|---|
| | | textured (%) | smooth | unknown (%) | | | local disease | metastatic disease | |
| UK | 23 | 23 (100) | 0 | 0 | 23 | 0 | 23 (100) | 0 | [14] |
| France | 19 | 18 (95) | 0 | 1 (5) | 18 | 2 (10) | 16 (84) | 3 (16) | [4] |
| worldwide | 173 | 171 (99) | 2 | 0 | nr | 9 (5) | 155 (89) | 8 (11) | [33] |
| worldwide | 63 | 26 (41) | 0 | 37 (59) | 18 | 4 (8) | 62 (98) | 1 (2) | [34] |
| USA | 34 | 4 (12) | 0 | 30 (88) | nr | nr | nr | nr | [18] |
| The Netherlands | 11 | 0 | 0 | 11 (100) | nr | nd | 8 (73) | 3 (27) | [6] |
| Australia/New Zealand | 55 | 55 (100) | 0 | 0 | nr | 4 (8) | 53 (96) | 2 (4) | [19] |
| worldwide | 130 | nr | nr | 130 (100) | nr | 5 (4) | 111 (85) | 19 (15) | [11] |

ooze from the implants into the surrounding tissue, it is probable that these tissues are also subject to infiltration by other active agents in the gels [46]. Of these, the various detected metals may directly induce oxidative stress and damage DNA [47]. Moreover, cells in any tissue chronically exposed to aromatic hydrocarbons may, due to the cytoplasmic ligand-activated transcription factor, the aryl hydrocarbon receptor (AHR), express several biotransforming cytochrome P450 enzymes (CYP 1Aa/2 and 1B1), which in turn may produce mutagenic reactive species from the aromatics [48]. These species may also induce oxidative stress and DNA damage. Breast cancer cells generally express high levels of the AHR, which participate in a large number of endogenous and anti-tumour functions, and studies on human and animals suggest that high AHR expression also concurs with inflammatory conditions [49,50]. One other decisive factor is that the AHR system, once activated by a number of ligands, also plays a key role in the production of inflammatory cytokines such as IL-17 [48]. Hence, AHR expression is a useful marker for both the formation of reactive species through CYP isoform-mediated biotransformation and the inflammatory process in primary and malignant cells. Whether the AHR is involved in BIA-ALCL pathogenesis remains to be seen, but given that BIA-ALCL is considered a malignancy of Th17/Th1 T-cell subsets, it is not improbable [29,51–53]. While a Th1/Th17 origin has been proposed, recent evidence is suggestive of a Th2 origin whereby eosinophils are seen in the involved capsules and tumour cells express the transcription factor GATA3 and producing IL13 are observed [52]. These latter data are supportive of the development of BIA-ALCL in the context of an allergic reaction.

Regardless of the mechanism(s) of disease pathogenesis, both systemic ALCL and BIA-ALCL share common oncogenic events; Jak1/STAT3 mutations are observed in systemic ALCL, ALK− and BIA-ALCL [54,55]. These proteins are also key factors downstream of hyperactive ALK activity in ALCL, ALK+, which suggests that they play a central role in ALCL [39,54–56]. However, alternative mechanisms for constitutive activation of STAT3 cannot be ruled out. The JAK/STAT3-targeted tyrosine kinase inhibitor (e.g. sunitinib) may nevertheless be considered an interesting strategy in the therapy of infiltrative BIA-ALCL. In contrast with primary cutaneous ALCL and a subgroup of systemic ALCL, ALK-associated with a favourable prognosis, there is no indication as to a DUSP22 rearrangement in BIA-ALCL. But in view of the small number of cases, further studies are required to determine whether BIA-ALCL is a new genetic subgroup of systemic ALCL, ALK− or a part of the still heterogeneous systemic ALCL, ALK− entity or indeed cutaneous ALCL due to its comparative indolent nature.

## 4. Are breast implants associated with other malignancies besides BIA-ALCL and other rare lymphomas?

Some rare cases of lymphomas besides BIA-ALCL have been reported in the context of breast implants, although whether these are attributable to the implants or are coincidental remains to be determined [57–61]. However, the risk of cancers, except lymphomas, developing due to breast implants has not, to date, been verified in any study. In 2007, Brinton

royalsocietypublishing.org/journal/rsob *Open Biol.* **9**: 190006

evaluated more than 13 000 women with breast implants and demonstrated slightly higher rates of cervical, vulvar, gastric and cerebral tumours, and leukaemias [62]. However, the differences observed may be attributable to different lifestyles between the two groups studied, rather than to breast implants. By contrast, Singh *et al.* did not find any increase in breast or other cancers, or any influence on systemic disorders such as lupus or Sjogren's syndrome upon analysis of 55 000 patients receiving Allergan implants [63].

# 5. Discussion

BIA-ALCL has been described in the scientific literature since 1997 and is considered uncommon, albeit almost exclusively in women with textured implants. To date, over 500 confirmed cases and 16 deaths have been described worldwide [64]. There is no consensus regarding the true incidence rate of BIA-ALCL as it varies worldwide and due to the high number of undetected BIA-ALCL or under-reported use of breast prostheses. Late-onset seroma and/or capsular contracture with unilateral breast swelling are some of the key symptoms and should be further investigated for BIA-ALCL by aspiration of seroma fluid with subsequent cytology, flow cytometry and CD30 testing. Complete removal of implants

as well as the capsule is curative in the majority of cases, particularly those diagnosed at stage 1A. When infiltration of the breast parenchyma, chest wall and/or lymph node involvement is seen, the prognosis is negatively impacted and adjuvant chemotherapy is required, particularly for non-resectable disease. The aetiology of BIA-ALCL remains to be determined but may be due to an infectious and/or toxic reaction coupled with allergy or autoimmunity. Based on these data, breast implants continue to be used in routine cosmetic and reconstructive breast surgery. What is clear is that all cases should be reported to their respective authorities and women should be informed of the risks, no matter how small, especially with regard to textured implants. It might also be necessary to halt the use of textured implants until more is understood about the pathogenesis of this disease [42]. The introduction of obligatory worldwide BIA-ALCL registers as well as reporting to the PROFLIE registry (www. thepsf.org/PROFILE) should improve the collection of data.

Data accessibility. This article has no additional data.

Competing interests. We declare we have no competing interests.

Funding. S.D.T. and L.K. are in receipt of an EU Marie Sklodowska Curie Innovative Training Network ALKATRAS, grant number 675712.

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
