## [Reviewer comments · Open Biology]

Review History

RSOB-19-0006.R0 (Original submission)

Review form: Reviewer 1

Recommendation

Reject – article is not of sufficient interest (we will consider a transfer to another journal)

Are each of the following suitable for general readers?

- a) **Title**
Yes
- b) **Summary**
Yes
- c) **Introduction**
Yes

Is the length of the paper justified?

Yes

Should the paper be seen by a specialist statistical reviewer?

No

Is it clear how to make all supporting data available?

Not Applicable

Is the supplementary material necessary; and if so is it adequate and clear?

No

Do you have any ethical concerns with this paper?

No

Comments to the Author

Although the article is well written, it represents only a review on this topic that has been already extensively reported in the literature. The question raised by the authors in the title of their manuscript is still under investigation and the authors also do not answer to the question. Moreover, authors support to halt the use of textured breast implants without a scientific explanation disregarding to prove that smooth implants are not involved in the pathogenesis of the lymphoma. The scope of scientific literature is to promote 1st level of evidence papers (multicenter randomized controlled trials) in this critical and worldwide discussed topic before coming to definitive conclusions and actions.

Review form: Reviewer 2

Recommendation

Accept with minor revision (please list in comments)

Are each of the following suitable for general readers?

- a) **Title**
Yes
- b) **Summary**
Yes
- c) **Introduction**
Yes

Is the length of the paper justified?

Yes

Should the paper be seen by a specialist statistical reviewer?

No

Is it clear how to make all supporting data available?

Yes

Is the supplementary material necessary; and if so is it adequate and clear?

Not Applicable

Do you have any ethical concerns with this paper?

No

Comments to the Author

The authors are quoting a risk for ALCL between 1:1,000 and 1:10,000 which is relatively high. no - one understands the true risk as denominator is unknown but in the UK the MHRA in 2019 guesstimates 1:24,000

Diagnosis and therapy: could the authors comment on role of imaging staging and role of MRI, PET-CT in diagnosed cases with BIA-ALCL.

Decision letter (RSOB-19-0006.R0)

28-Feb-2019

Dear Dr Turner,

We are pleased to inform you that your manuscript RSOB-19-0006 entitled "Is Breast Implant-Associated Anaplastic Large Cell Lymphoma (BIA-ALCL) a Hazard of Breast Implant Surgery?" has been accepted by the Editor for publication in Open Biology. The reviewer(s) have recommended publication, but also suggest some minor revisions to your manuscript. Therefore, we invite you to respond to the reviewer(s)' comments and revise your manuscript.

Please submit the revised version of your manuscript within 14 days. If you do not think you will be able to meet this date please let us know immediately and we can extend this deadline for you.

1) A text file of the manuscript (doc, txt, rtf or tex), including the references, tables (including

captions) and figure captions. Please remove any tracked changes from the text before submission. PDF files are not an accepted format for the "Main Document".

2) A separate electronic file of each figure (tiff, EPS or print-quality PDF preferred). The format should be produced directly from original creation package, or original software format. Please note that PowerPoint files are not accepted.

3) Electronic supplementary material: this should be contained in a separate file from the main text and meet our ESM criteria (see <http://royalsocietypublishing.org/instructions-authors#question5>). All supplementary materials accompanying an accepted article will be treated as in their final form. They will be published alongside the paper on the journal website and posted on the online figshare repository. Files on figshare will be made available approximately one week before the accompanying article so that the supplementary material can be attributed a unique DOI.

Online supplementary material will also carry the title and description provided during submission, so please ensure these are accurate and informative. Note that the Royal Society will not edit or typeset supplementary material and it will be hosted as provided. Please ensure that the supplementary material includes the paper details (authors, title, journal name, article DOI). Your article DOI will be 10.1098/rsob.2016[last 4 digits of e.g. 10.1098/rsob.20160049].

4) A media summary: a short non-technical summary (up to 100 words) of the key findings/importance of your manuscript. Please try to write in simple English, avoid jargon, explain the importance of the topic, outline the main implications and describe why this topic is newsworthy.

Images

Data-Sharing

It is a condition of publication that data supporting your paper are made available. Data should be made available either in the electronic supplementary material or through an appropriate repository. Details of how to access data should be included in your paper. Please see <http://royalsocietypublishing.org/site/authors/policy.xhtml#question6> for more details.

Data accessibility section

Sincerely,

The Open Biology Team
<mailto:openbiology@royalsociety.org>

Reviewer(s)' Comments to Author:

Referee: 1

Comments to the Author(s)

Although the article is well written, it represents only a review on this topic that has been already extensively reported in the literature. The question raised by the authors in the title of their manuscript is still under investigation and the authors also do not answer to the question. Moreover, authors support to halt the use of textured breast implants without a scientific explanation disregarding to prove that smooth implants are not involved in the pathogenesis of the lymphoma. The scope of scientific literature is to promote 1st level of evidence papers (multicenter randomized controlled trials) in this critical and worldwide discussed topic before coming to definitive conclusions and actions.

Referee: 2

Comments to the Author(s)

The authors are quoting a risk for ALCL between 1:1,000 and 1:10,000 which is relatively high. no - one understands the true risk as denominator is unknown but in the UK the MHRA in 2019 guesimates 1:24,000

Diagnosis and therapy: could the authors comment on role of imaging staging and role of MRI, PET-CT in diagnosed cases with BIA-ALCL.

Author's Response to Decision Letter for (RSOB-19-0006.R0)

See Appendix A.

Decision letter (RSOB-19-0006.R1)

12-Mar-2019

Dear Dr Turner

We are pleased to inform you that your manuscript entitled "Is Breast Implant-Associated Anaplastic Large Cell Lymphoma (BIA-ALCL) a Hazard of Breast Implant Surgery?" has been accepted by the Editor for publication in Open Biology.

Article processing charge

Please note that the article processing charge is immediately payable. A separate email will be sent out shortly to confirm the charge due. The preferred payment method is by credit card; however, other payment options are available.

Sincerely,

The Open Biology Team
mailto: openbiology@royalsociety.org

Appendix A

DEPARTMENT OF PATHOLOGY
MEDICAL UNIVERSITY OF VIENNA

Prof. Dr Lukas Kenner
Deputy Director, Dept. of Pathology

Medical University of Vienna
Department of Pathology
Waehringer Guertel 18-20
1090 Vienna, Austria

Phone: 43 (0)1 40400-36590
lukas.kenner@meduniwien.ac.at
www.meduniwien.ac.at

Vienna, 04 March 2019

Re: RSOB-19-0006, "Is Breast Implant-Associated Anaplastic Large Cell Lymphoma (BIA-ALCL) a Hazard of Breast Implant Surgery?"

Dear Open Biology Team

We thank the reviewers for their comments and have replied to these as detailed below:

Referee: 1

Comments to the Author(s)

Although the article is well written, it represents only a review on this topic that has been already extensively reported in the literature. The question raised by the authors in the title of their manuscript is still under investigation and the authors also do not answer to the question. Moreover, authors support to halt the use of textured breast implants without a scientific explanation disregarding to prove that smooth implants are not involved in the pathogenesis of the lymphoma. The scope of scientific literature is to promote 1st level of evidence papers (multicenter randomized controlled trials) in this critical and worldwide discussed topic before coming to definitive conclusions and actions.

We are grateful to this reviewer for their comments and whilst we appreciate there are a number of reviews on this topic, ours is succinct, easy to read and will help to disseminate information on BIA-ALCL to a broader audience. Indeed, the title of the paper is a question that needs to be answered and that is why we have posed it. We present current evidence in an easy to absorb manner to allow the reader to come to their own conclusions. We do not suggest that all use of textured implants should be stopped, we merely suggest that this might be necessary as further evidence arises - "It might also be necessary to halt the use of textured implants until more is understood about the pathogenesis of this disease".

Referee: 2

Comments to the Author(s)

The authors are quoting a risk for ALCL between 1:1,000 and 1:10,000 which is relatively high. no-one understands the true risk as denominator is unknown but in the UK the MHRA in 2019 guesstimates 1:24,000

We have clarified the incidences based on the country in which they were reported and whether the rate cited is based on women with textured or all implants:

Current expert opinion puts the risk of ALCL at between 1:2,832 for women with polyurethane implants in Australia and New Zealand to 1:30,000 women with any textured implant in the USA, with the current UK risk estimated to be 1:24,000, a figure based on all breast implants sold.

Diagnosis and therapy: could the authors comment on role of imaging staging and role of MRI, PET-CT in diagnosed cases with BIA-ALCL.

We have added the following to the text:

Late onset seromas (> 1 year) should be investigated by ultrasound of the breasts, chest wall and regional lymph nodes. When ultrasound is inconclusive, magnetic resonance imaging or computed tomography scans may assist with the diagnosis of soft tissue masses (8). Ultrasound-guided aspiration of seroma fluid with subsequent cytology, flow cytometry and analysis of the cells, primarily for CD30 expression and negativity for ALK helps to establish the diagnosis.

Sincerely,

Prof. Dr. Lukas Kenner
Head of the Department of Experimental Pathology and Laboratory Animal Pathology, Medical University and University of Veterinary Medicine Vienna
Deputy Director: Department of Pathology
Director: Christian Doppler Laboratory for Applied Metabolomics